# Transcriptomic Profiling of Peripheral Edge of Lesions to Elucidate the Pathogenesis of Psoriasis Vulgaris

**DOI:** 10.3390/ijms23094983

**Published:** 2022-04-30

**Authors:** Suphagan Boonpethkaew, Jitlada Meephansan, Onjira Jumlongpim, Pattarin Tangtanatakul, Wipasiri Soonthornchai, Jongkonnee Wongpiyabovorn, Ratchanee Vipanurat, Mayumi Komine

**Affiliations:** 1Division of Dermatology, Chulabhorn International College of Medicine, Rangsit Campus, Thammasat University, Klong Luang, Pathum Thani 12120, Thailand; suphagan.b@gmail.com (S.B.); onjijum@gmail.com (O.J.); 2Department of Transfusion Medicine and Clinical Microbiology, Faculty of Allied Health Sciences, Chulalongkorn University, Bangkok 10330, Thailand; pattarin.t@chula.ac.th; 3School of Science, University of Phayao, Phayao 56000, Thailand; wipasiri39@hotmail.com; 4Center of Excellence in Immunology and Immune-Mediated Disease, Department of Microbiology, Faculty of Medicine, Chulalongkorn University, Bangkok 10330, Thailand; jongkonneew@gmail.com; 5Division of Dermatology, Department of Medicine, Rajavithi Hospital, Ministry of Public Health, Bangkok 10400, Thailand; toffytoffy@hotmail.com; 6Department of Dermatology, Jichi Medical University, 3311-1 Yakushiji, Shimotsuke, Tochigi 329-0498, Japan; mkomine12@jichi.ac.jp

**Keywords:** RNA sequencing, inflammation, cytokines, chronic skin diseases, gene

## Abstract

Elucidating transcriptome in the peripheral edge of the lesional (PE) skin could provide a better understanding of the molecules or signalings that intensify inflammation in the PE skin. Full-thickness biopsies of PE skin and uninvolved (UN) skin were obtained from psoriasis patients for RNA-seq. Several potential differentially expressed genes (DEGs) in the PE skin compared to those in the UN skin were identified. These DEGs enhanced functions such as angiogenesis, growth of epithelial tissue, chemotaxis and homing of cells, growth of connective tissues, and degranulation of myeloid cells beneath the PE skin. Moreover, the canonical pathways of IL-17A, IL-6, and IL-22 signaling were enriched by the DEGs. Finally, we proposed that inflammation in the PE skin might be driven by the IL-36/TLR9 axis or IL-6/Th17 axis and potentiated by IL-36α, IL-36γ, IL-17C, IL-8, S100A7, S100A8, S100A9, S100A15, SERPINB4, and hBD-2. Along with IL-36α, IL-17C, and IκBζ, ROCK2 could be an equally important factor in the pathogenesis of psoriasis, which may involve self-sustaining circuits between innate and adaptive immune responses via regulation of IL-36α and IL-36γ expression. Our finding provides new insight into signaling pathways in PE skin, which could lead to the discovery of new psoriasis targets.

## 1. Introduction

Psoriasis is a chronic skin disease influenced by the interplay of genetic predisposition, environmental triggers, and immune response. The classic feature of chronic plaque or psoriasis vulgaris is a well-demarcated salmon-pink plaque covered with silvery scales [1]. In addition to psoriasis vulgaris, inverse, guttate, pustular, and erythrodermic forms have been reported as the different phenotypes of psoriasis, which may be associated with comorbidities such as arthritis, metabolic syndrome, and cardiovascular diseases [1,2,3]. Various genes involved in antigen presentation (*HLA-Cw6*), skin barrier formation, innate immunity, and adaptive immunity have been identified as psoriasis-susceptible genes that increase the risk of developing the disease [4,5]. Individuals who carry these susceptible genes will be more likely to develop psoriasis under the influence of environmental factors, which initiate an inflammatory response leading to psoriasis phenotypes [1,5,6,7].

In the initial phase, injured keratinocytes release self-DNA/RNA and cathelicidin (LL-37), which form an LL-37/self-RNA/DNA complex that activates plasmacytoid dendritic cells (pDCs) via Toll-like receptor (TLR)7/9. The activated pDCs produce interferon (IFN)-α and -β to activate myeloid dendritic cells (mDCs). In addition to pDC-derived IFN-α, activated keratinocyte-derived interleukin (IL)-1β, IL-6, and tumor necrosis factor (TNF) can mature and activate myeloid dendritic cells (mDCs) to produce IL-23 [5,8]. Subsequently, the mDC- derived IL-23 drives the differentiation of T helper (Th)17 and T cytotoxic (c)17 cells. Both Th17 and Tc17 cells provide a substantial source of IL-17A and IL-22, which are the key cytokines for antimicrobial peptide (AMP) induction and keratinocyte (KC) activation to produce T cell- and neutrophil-chemokines and promote KC proliferation, ultimately resulting in the typical features of psoriasis. Thus, the IL-23/Th17 axis has been suggested as the main psoriasis driver [5,7,8]. In addition to the three main cells (DCs, T cells, and KCs), neutrophils, innate lymphoid cells (ILCs), Langerhans cells, macrophages, and natural killer cells could contribute to the pathogenesis of psoriasis [5,8]. Currently, there are many effective biologics available for treatment, including TNF inhibitors, IL-12 and IL-23 blockers, and anti-IL-17. However, the adverse effects of these drugs necessitate long-term monitoring, and there is currently no therapeutic option to cure psoriasis due to the complex and not yet fully elucidated pathogenesis [1,9,10].

The first RNA-sequencing (RNA-seq) analysis in psoriatic tissue was performed in 2012, which identified psoriasis-associated differentially expressed genes (DEGs), and has facilitated further studies to gain a better understanding of psoriasis pathogenesis [11]. Although the number of genes associated with psoriasis has been increasing, most of these have been identified from lesional (LS) skin samples [11,12,13,14,15,16,17,18,19,20,21]. Recently, transcriptomic studies in psoriasis have been focusing on more cell-specific or single-cell RNA-seq approaches; these approaches provide insight into psoriasis pathogenesis [22,23,24]. However, the peripheral edge of lesional (PE) skin RNA-seq is still unexplored. In the active edge lesion, psoriatic vascular change precedes epidermal and immune cell changes [25,26]. Furthermore, the interaction between dendritic cells and T cells is thought to initiate early inflammation and plaque propagation beneath the perilesional (PR) skin, a normal-appearing skin adjacent to the PE skin [27,28]. Hence, these observations suggested that the skin may have some molecules or signaling pathways that focus on intensifying inflammation in the transitional zone, which might contribute to plaque propagation. We here present the first transcriptomic profile of the PE skin along with a proposed network. These findings could provide new insight to expand the molecular network contributing to the pathogenesis of psoriasis toward identifying new therapeutic options for patients with psoriasis vulgaris.

## 2. Results

### 2.1. DEGs in the PE Skin

A total of 1202 DEGs were identified. Of these, 653 (54%) were upregulated, and 549 (46%) were downregulated in PE skin compared to the uninvolved (UN) skin (Figure 1). A heatmap of the differentially regulated transcripts between the PE and UN skin is shown in Appendix A. The top 50 upregulated and downregulated DEGs according to the log2FC value are listed in Table 1. The top 100 DEGs are shown in Appendix A.

### 2.2. DEG Analysis

#### 2.2.1. Disease Associations and Biological Functions

The 1202 DEGs were analyzed for downstream effects in terms of associated diseases; among them, 135 genes have previously been reported to be associated with psoriasis based on the Ingenuity Knowledge Base, 86 of which showed upregulated in the PE skin. Thus, we hypothesized that the proteins encoded by these 86 DEGs might be involved in inflammation in the PE skin to contribute to psoriasis (Figure 1 and Appendix A). Furthermore, the 1202 DEGs were analyzed for their functions. These DEGs enhanced functions are to enrich angiogenesis (*EDNRB*, *LAMA2, S100A8, S100A9, SPINK5, TNFRSF1A*, and *VAV3*), growth of epithelial tissue (*ADAM17, KRT16, IKBKB, IL36G, ITGB1, RICTOR*, and *XDH*), chemotaxis and homing of cells (*A2M, C10orf99, DEFB4A/DEFB4B, GFRA1, PLEC, S100A7*, and *S100A7A)*, growth of connective tissue (*ADAR, ELN, IL36A, KLK6, PLAT, STAT1*, and *STAT3*) and degranulation of myeloid cells (*CD59, CXCL8, DSC1, DSG1, LCN2*, and *SERPINB3*). The list of all DEGs in each function is provided in Appendix A.

In addition, the upregulated- and downregulated DEGs were analyzed separately as pathway and function enrichments using Metascape. Notably, cornified envelope formation, VEGFA-VEGFR2 signaling, interferon signaling, and neutrophil degranulation were enriched by upregulated DEGs (Figure 2A), whereas epithelial differentiation was enriched by downregulated DEGs (Figure 2B) such as *KRT10, KRT2, LCE1C, LCE1D, LCE2B*, and *LORICRIN*. The list of all DEGs in each DEG-enriched pathway and function is provided in Appendix A. We reviewed and selected the potential DEGs based on their biofunctions and psoriasis associations; they are shown in Table 2 and Figure 3. Moreover, we compared and mapped PE vs. UN skin DEGs from previous studies [11,12,13,14,22,23,24,29,30]; the details are provided in Appendix A.

#### 2.2.2. Canonical Pathways

We analyzed the known canonical pathways enriched by the DEGs of PE skin and the activity prediction of these pathways based on the QIAGEN Knowledge Base. To include both significant and highly predicted activity pathways, we determined the threshold of minimum significance (*p*-value of overlap > 0.05 or −log (*p*-value) > 1.3) and absolute z-score (|z-score| ≥ 2). Only seven canonical pathways are shown in Figure 4A,B. It is noteworthy that the role of IL-17A in psoriasis, eukaryotic initiation factor 2 (eIF2) signaling, and the coronavirus pathogenesis pathway had the top three highest ratios (mapped DEGs divided by total molecules in a given pathway = 0.36, 0.30, and 0.25, respectively). Moreover, these three pathways are involved in the stress response and inflammation, especially the coronavirus pathogenesis pathway, the hallmark of which is the cytokine storm [31]. The ratios of all seven pathways are listed in Appendix A.

To explore the role of cytokine signaling in the PE skin in further detail, we analyzed the canonical pathway of cytokines, with a −log (*p*-value) > 1.3 indicating significance. A total of 11 cytokine signaling pathways were among the DEG-enriched pathways (Figure 4C,D), including the IL-8, IL-17A, IL-6, C-C motif chemokine receptor 3(CCR3), IL-15, granulocyte-macrophage colony-stimulating factor (GM-CSF), IL-22, and formyl-methionyl-leucyl-phenylalanine (fMLP) signaling pathways.

#### 2.2.3. Upstream Regulators

We further identified the upstream regulators of the DEGs in the PE skin. We used the upstream analysis function according to a statistically significant overlap between the dataset genes and the genes that are regulated by a potential regulator (*p*-value overlap < 0.05) and |z-score| ≥ 2. The lists of potential regulators with their predicted activity are provided in Appendix A. Among the potential regulators, seven molecules were identified as DEGs, including *CCND1, DDX21, IL36A, KDM5A, NUPR1, RICTOR*, and *STAT1*. Some relationships between the potential regulators and their target DEGs are shown in Figure 5 and Appendix A.

#### 2.2.4. Regulator Effect

Next, we established a causal hypothesis between upstream regulators and their downstream effects using the regulator effect function to correlate the significant upstream regulators with their downstream functions. The useful predicted networks were prioritized and ranked according to their consistency scores. The two highest consistency score networks are shown in Figure 6. Notably, IL-36α, IL-17C, nuclear factor-kappa B inhibitor zeta (IκBζ), and Rho-associated kinase 2 (ROCK2) emerged as potentially important regulators in the network.

#### 2.2.5. Validation of Selected DEGs by Multiplex Real-Time PCR

Finally, to validate the expression of DEGs such as *SERPINB4, IL36A*, *PLSCR1, CXCL8, DEFB4A/B, DEFB103A/B, S100A7, S100A7A, S100A8*, and *S100A9*, we performed quantitative multiplex Real-Time PCR. Furthermore, we compared the mRNA expression of upstream regulators IL-17C, IκBζ, and ROCK2 in the PE skin to those in the UN skin. The results are presented in Figure 7.

## 3. Discussion

To the best of our knowledge, this is the first study to establish the transcriptomic profile at the edge of psoriatic lesions. We revealed that the increased functions in the PE skin were involved in angiogenesis, growth of epithelial tissue, chemotaxis and homing of cells, growth of connective tissue, and the degranulation of myeloid cells. The upregulated DEGs were enriched in functions such as cornified envelope formation, VEGFA-VEGFR2 signaling, interferon signaling, and neutrophil degranulation. Moreover, we showed that the genes involved in epithelial differentiation were downregulated in the PE skin compared with the UN skin, which might limit the normal differentiation capacity of KC and contribute to plaque formation. We discussed some potential DEGs that were related to AMPs, angiogenesis, autoantigens, and chemotaxis that might play a pivotal role in the pathogenesis of PE skin.

AMPs activate the innate immune response to create an inflammatory environment in the skin through antimicrobial, chemotactic, angiogenic, and proinflammatory properties [32,33]. We identified various AMP-coding genes that were upregulated in the PE skin. We confirmed that *DEFB4A/B* mRNA was significantly upregulated in PE skin and its fold change was the highest among the AMP-coding genes. It was previously reported to be expressed in psoriatic plaques [34,35] and more highly in lesional skin than in UN skin [36]. Its protein, human β-defensin 2 (hBD-2), promotes pDCs to uptake self-DNA and secrete IFN-α in a TLR-9–dependent manner [37]. Moreover, we confirmed that *S100A7, S100A7A, S100A8*, and *S100A9* mRNA were significantly upregulated in PE skin compared to those in UN skin. Interestingly, among the psoriasis-associated S100 protein-coding genes, *S100A7A* (also known as *S100A15*) was the most upregulated gene in the PE skin. Its protein koebnerisin is expressed in psoriatic lesions along with psoriasin (encoded by *S100A7*), S100A8, and S100A9 [38,39] by induction of psoriatic-derived Th17/Th22 cytokines (IL-17, IL-22, and TNF-α) [40,41,42]. Of particular interest, apart from their antimicrobial properties of them [33,43], both koebnerisin and psoriasin have been proposed to be responsible for the inflammation priming of KCs [41], as well as S100A8 and S100A9 also function as proangiogenic factors [39]. However, further study is needed to determine their role in the PE skin, in particular the very-high-fold change of *S100A7A*.

We also identified the upregulation of *SERPINB3* and *SERPINB4* in the PE skin and confirmed that *SERPINB4* was significantly upregulated in the PE skin. Their proteins, serpin family B members 3 and 4, were reported to be expressed in psoriatic lesions [44] by induction of IL-17 and IL-22 [45]. Pso p27 antigen, a proteolyzed product of SERPINB4 and SERPINB3 [46,47], was detected in the PE skin, which is considered to be equivalent to its detection on the lesion, but was not detected on the UN skin [48]. The pso p27 forms an immune complex with anti-pso p27 to activate the complement cascade at the psoriatic scale [49]. However, the roles of SERPINB3 and SERPINB4 as antigens in psoriasis are controversial and require further investigation, especially *SERPINB4*, which was highly upregulated in the PE skin.

We further used the canonical pathway, regulator analysis, and regulator effect functions of QIAGEN’s IPA tool to generate the network among the molecules. The proposed pathogenic network is schematically displayed in Figure 8. We found that *IL36A* and *IL36G* expression were upregulated in the PE skin and then confirmed that *IL36A* expression was significantly upregulated in the PE skin. Their proteins IL-36α and IL-36γ are expressed in psoriatic plaque [50,51,52], and their mRNA levels were reported to be increased in KCs after injury [53,54]. Moreover, IL-36α and IL-36γ alone or synergistically with IL-17A could amplify their own mRNA and proteins in KCs [55]. We identified and confirmed the significant upregulation of *PLSCR1* (encoding a TLR-9 transporter) in the PE skin. A previous RNA-seq analysis of whole blood from psoriasis patients implicated the pDC-mediated release of IFN-α after induction by IL-36 in a TLR-9-dependent manner, evidenced by the upregulation of *PLSCR1* in pDCs [56]. The pDC-derived IFN-α promotes mDC maturation and function, which amplifies and sustains T cells for the psoriatic inflammatory response [7,8]. Therefore, we suggest that the IL-36/TLR-9 axis, which upregulates IFN-α production, might also drive psoriatic inflammation in the PE skin (Figure 8).

IL-36α and IL-36γ also activate mDCs to mature and secrete IL-6 [57]. Dermal DC-derived IL-6 then drives Th17 and Th22 cell differentiation [7,58]. Furthermore, IL-6 is also secreted by KCs after IFN-γ treatment via the suppression of miR-149, which amplifies skin inflammation [59]. Consistently, we found that the DEGs in the PE skin were enriched in the IL-6 canonical pathway, suggesting that the IL-6/Th17 axis might be another axis driving psoriatic inflammation in the PE skin (Figure 8). However, despite the lack of IL-6, KCs expressed various additional proinflammatory cytokines after the transient suppression of psoriasiform skin manifestation, delineating the ineffective treatment of psoriasis plaque by IL-6 blockage [60]. Our results showed that the DEGs were enriched in the IL-17A and IL-22 canonical pathways and that IL-22 was an activated potential regulator, thus suggesting the crucial roles of IL-17A and IL-22 in the PE skin. IL-17A and IL-22 potentiate the psoriatic milieu by inducing AMPs, recruiting other immune cells, and increasing KC proliferation [7,8]. In addition, a synergistic effect of the Th17 cytokines IL-17A and IL-22 induces the expression of IL-36α and IL-36γ mRNA in KCs [50,61]. IL-22, in particular, promotes KC proliferation and suppresses KC terminal differentiation [42]. Previous studies comparing LS skin to healthy skin suggested that IL-22 functions better when the levels of soluble scavenging receptor IL-22 binding protein (encoded by *IL22RA2*) are decreased. However, we could not identify the alteration of *IL22RA2* as DEG of PE vs. UN skin. Furthermore, serum IL-22 level was higher in psoriasis patients compared with healthy individuals and was also positively correlated with disease severity [42,62].

Considering IL-36, IL-36α was identified as an activated potential regulator in the PE skin, which induces the expression of *IL36G, IL17C, S100A9*, and *NFKBIZ* in KCs [63]. Notably, the mouse skin showed enhanced chemokine expression, leukocyte infiltration, and acanthosis after the injection of IL-36α [57]. Moreover, imiquimod induced much milder psoriasiform dermatitis in *Il36a*^−/−^ mice compared with that induced in wild-type mice [64].

IL-17C was identified as an activated potential regulator. We established that its mRNA expression was not significantly upregulated in the PE skin because similar relative expression levels were also detected in the UN skin. These results suggest that IL-17C plays a role during the early stages of psoriasis inflammation or inflammatory priming for plaque formation. A previous report demonstrated that IL-17C transgenic mice developed psoriasiform skin. Moreover, IL-17C and its mRNA significantly increase in UN and LS skin compared to healthy skin [65]. IL-17C is mainly secreted from KCs and not leukocytes [66]. It has also been detected in psoriatic plaques at a higher concentration than IL-17A [65,67]. IL-17C stimulates the release of TNF-α from monocytes [68] and also simulates neutrophil migration, especially in synergy with TNF-α [67]. IL-17C could amplify itself in KCs, especially in synergy with TNF-α [65,67]. In addition, IL-36α and IL-36γ alone or synergistically with IL-17A increased IL-17C levels from KCs [55]. Moreover, IL-17C induced the expression of *S100A15, S100A7, S100A8, S100A9, DEFB4A, CXCL8, IL36G, NFKBIZ*, and *TNIP3* in KCs [66,67]. In particular, IL-17C synergistically with TNF-α induced the expression of *S100A7, S100A8, S100A9, LCN2* (coding lipocalin-2), *CXCL8*, *IL36G*, and *DEFB4* in KCs [65].

IκBζ was identified as an activated potential regulator in the PE skin network. Moreover, we also observed that *NFKBIZ* mRNA was significantly upregulated in the PE skin. A recent study reported that *NFKBIZ* was co-expressed with *IL36G* in the stratum spinosum of psoriasis epidermis [24]. IκBζ is a nuclear transcriptional cofactor that plays a role as the mediator of the expression of IL-36α-induced genes, including *S100A7, S100A8, S100A9, LCN2, DEFB4, IL36G, SERPINB4*, and *CXCL8* [63], IL-17A-induced genes, including *DEFB4, S100A7, NFKBIZ*, and *LCN2* [69,70], and IL-17A/TNF-α-induced gene, including *DEFB4, LCN2, CXCL8, IL17C, NFKBIZ*, and *IL36G* [69,70,71,72] in KCs.

We identified significant upregulation of *CXCL8* in the PE skin. It encodes IL-8, which attracts T lymphocytes to the psoriasis plaque [73], promotes angiogenesis [74,75], and promotes epidermal proliferation [76]. Interestingly, IL-8 is secreted by accumulated neutrophils (Munro’s microabscess) and is the chemokine for other neutrophils to aggregate into the plaque in an autocrine manner [77]. Moreover, IL-8 is secreted from KCs after scratching [54] or the induction of IL-22 or induction of IL-36α and IL-36γ, especially in synergy with IL-17A [55,57,61,78]. Particularly, KC-derived IL-8 is capable of neutrophil chemotaxis, but this capacity was reduced when miR-146a, an IL-17 counteractant, which is expressed in psoriasis plaque, was overexpressed [79].

We identified that ROCK2, as an activated potential regulator, could be equally important as IL-36α, IL-17C, and IκBζ in the network. ROCK2 is located in the nucleus and cytoplasm of the human T- cell [80]. It plays an important role in keratinocyte differentiation [81] as well as in angiogenesis [82]. Moreover, ROCK2 was found to regulate CD4^+^ T cells to secrete IL-17 [83]. These processes are important for the formation of psoriasis plaques [7]. In 2017, the activity of ROCK2 was studied in moderately severe plaque psoriasis patients using selective oral ROCK2 inhibitors; 71% of the patients achieved a reduction of 50% in the psoriasis area and severity index (PASI 50) at 12 weeks, peripheral blood IL-17 and IL-23 significantly decreased, and ROCK2 staining decreased with decreasing epidermal thickness [84]. As the predicted regulator in the PE skin, ROCK2 was associated with the upregulation of nine DEGs (*CXCL8, DSC1, DSC2, DSG1, DSG3, SPINK5, PI3, S100A7*, and *S100A8*) (Figure 5). We further investigated the *ROCK2* mRNA expression in PE skin and compared it to those in the UN skin. Our results showed that *ROCK2* mRNA was significantly downregulated. However, activation of the ROCK2 kinase protein depends on post-translational modification steps. Previous reports suggest that activation of ROCK2 kinase needs phosphorylation [85,86]. Interestingly, its structure is capable of self-phosphorylation (auto-phosphorylation) [87]. Thus, we suggest that the downregulation of *ROCK2* mRNA in PE skin might be the result of autoregulation overfunction. In addition, ROCK2 kinase has been proposed to cooperate with phosphorylated signal transducer and activator of transcription 3 (STAT3) to regulate Th17 function [80]. Due to the complexity of psoriasis, further investigation of *ROCK2* mRNA expression in the PE skin compared to the mature lesion or healthy skin could provide new information toward a better understanding of the mechanism of oral ROCK2 inhibitor action and the role of ROCK2 in the pathogenesis of psoriasis.

Finally, based on our results and these previous findings, we propose that IL-36α, IL-36γ, IκBζ, IL-17C, IL-8, S100A7, S100A8, S100A9, S100A15, SERPINB4, hBD-2, and ROCK2 potentiate inflammation in the PE skin and that IL-36α and IL-36γ may run a self-sustaining circuit between the innate and adaptive immune response, creating a psoriatic milieu in the PE skin (Figure 8).

The limitation of this study is the small sample size. Ideally, more samples of a larger size need to be evaluated to determine the PE skin transcriptomic profile.

In conclusion, this study identified a set of upregulated DEGs, including *PLSCR1, IL36A, CXCL8, S100A7, S100A8, S100A9, S100A15, DEFB4A/B*, and *SERPINB4*, and the molecules that may regulate these DEGs including IL-17C, IL-36α, IκBζ, and ROCK2. These DEGs and upstream regulators may potentiate the inflammation in PE skin as the proposed network might be different from that seen in the LS skin. This study provides new insight into the molecular mechanisms occurring in the PE skin and could lead to the identification of specific molecules that could serve as therapeutic targets in psoriasis vulgaris. Further, it would be helpful to compare these molecules with those observed in the center of LS skin, normal-appearing skin in patients with psoriasis, and healthy skin, to further confirm changes in the levels of other DEGs and regulators of interest at both the mRNA and protein levels, and to further examine therapeutic modalities that might target these molecules by comparing the before and after treatment transcript levels in the PE skin. These insights could provide a better understanding of the pathogenesis of psoriasis and lead to the discovery of novel therapeutic strategies.

## 4. Materials and Methods

### 4.1. Patients

We enrolled voluntary patients with plaque-type psoriasis (aged ≥18 years) with moderate-to-severe psoriasis [Psoriasis Area and Severity Index (PASI) score ≥ 10] diagnosed by either classical clinical features or a confirmed pathological examination. Patients who had been treated with topical agents (steroids, vitamin D analogs) within 2 weeks or had undergone phototherapy and systemic medication (prednisolone, methotrexate, biologics) within 4 weeks of the date of tissue sampling were excluded. Based on previous RNA-seq profiling studies in human psoriasis skin, three samples were deemed to be adequate for analysis [11,88]; therefore, three eligible patients were included in this study. The PASI scores of the three patients are shown in Appendix A. All three patients were informed of all study protocols and signed the consent forms. The protocol was designed in accordance with the principles of the Declaration of Helsinki and was approved by the ethics committee of Thammasat University, Thailand.

### 4.2. Tissue Sampling

Full-thickness skin biopsies were collected from the three patients using a 6 mm punch biopsy under local anesthesia (2% lidocaine with adrenaline). The PE skin was taken at the edge of the plaque (Figure 9), and UN skin was collected from phenotypically normal skin a minimum of 10 cm away from the plaque from the same patient. The biopsies were preserved in RNAlater (ThermoFisher, AMBION, Austin, TX, USA) at −80 °C until gene expression profiling.

### 4.3. High-Throughput Next-Generation Sequencing (NGS)

Total RNA was extracted from each skin biopsy of the volunteers (PE skin, *n* = 3, UN skin, *n* = 2) using TRIzol Reagent (Ambion, Austin, TX, USA). NGS was performed by Macrogen (Seoul, Korea). Briefly, the purity and structural integrity of RNA were analyzed using an Agilent 2100 Bioanalyzer (RNA integrity number > 7). Sequence libraries were constructed using the SMARTer Universal Low Input RNA Kit and TruSeq RNA Sample Prep Kit v2 (pair-end). NGS was performed using the NovaSeq 6000 S4 Reagent Kit on the NovaSeq 6000 System (Illumina Inc., San Diego, CA, USA) with 100-bp paired-end reads. The raw data generated in this study have been deposited in the NCBI Gene Expression Omnibus (GEO; http://www.ncbi.nlm.nih.gov/geo, accessed on 8 September 2021) and are accessible through the GEO Series accession number GSE183732.

### 4.4. Sequencing Data Analysis

Basic data analysis was conducted by Macrogen. Briefly, overall read qualities, total bases, total reads, guanine-cytosine content (%GC content), and basic statistics were calculated. Trimmed reads were mapped to the reference genome using HISAT2, a splice-aware aligner. The transcripts were assembled using StringTie with aligned reads. Expression profiles were repeated as reading counts and normalized based on the transcript length and depth of coverage. The fragments per kilobase of transcript per million mapped reads (FPKM) value was used as a normalization value. Furthermore, we used the read counts to statistically analyze the expression profile with edgeR Bioconductor statistical library version 3.13 [89,90] on R Studio [91]. The DEGs between the PE skin and UN skin were determined according to *p* < 0.01 (0.026 ≤ FDR ≤ 0.194) and fold change (FC) ratio (|log2FC|) ≥ 1.5.

### 4.5. DEGs Analysis

All DEGs with expression profiles were uploaded into QIAGEN’s Ingenuity Pathway Analysis (IPA) software (QIAGEN Redwood City, CA, USA; www.qiagen.com/ingenuity, accessed on 1 October 2021) using the “core analysis” of DEGs. Disease and biofunction, canonical pathways, upstream regulators, and regulator effects of the DEGs were analyzed based on the Ingenuity Knowledge Base. Pathway and function enrichment analysis was additionally analyzed using Metascape Online Tool (https://metascape.org, accessed on 1 October 2021) [92].

### 4.6. Quantitative Multiplex Real-Time PCR (Multiplex qPCR)

For validation experiments, UN and PE skin samples were biopsied from moderate to severe psoriasis vulgaris patients. RNA was extracted and reverse transcribed to complementary DNA using the ImProm-II^TM^ Reverse Transcription Kit (Promega Corp., Madison, WI, USA). TagMan^®^ Gene Expression Assay predesigned qPCR primers of the selected DEGs (*IL36A, CXCL8, PLSCR1, SERPINB4, DEFB4A/B, DEFB103A/B, S100A7, S100A7A, S100A8*, and *S100A9*), upstream regulators (*IL17C, NFKBIZ*, and *ROCK2*), and of housekeeping gene (*GAPDH*) for qPCR were purchased from Gene Plus Co., Ltd. (ThermoFisher Scientific, West Sacramento, CA, USA). Their assay IDs are listed in Appendix A. The multiplex qPCR was performed using the QuantiStudio^TM^ 6 Flex Real-Time PCR system (Applied Biosystem). A statistical difference in the mean of ∆Ct (targeted gene vs *GAPDH*) between the UN and PE skin groups was calculated by a paired *t*-test. Graphs created by GraphPad Prism 9.3.1 were shown for relative mRNA expression using the 2^−∆Ct^ method and for fold change using the 2^−∆∆Ct^ method [93].

## Figures and Tables

**Figure 1 ijms-23-04983-f001:**
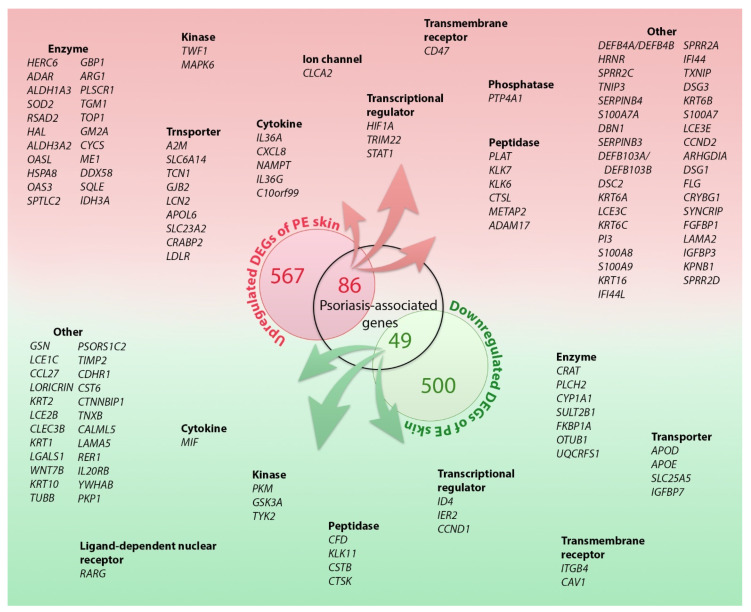
The differentially expressed genes (DEGs). A total of l202 genes was identified as the DEGs in the peripheral edge of the lesional (PE) skin vs. uninvolved (UN) skin. Of these, 653 (54%) were upregulated and 549 (46%) were downregulated. Among these DEGs, 135 DEGs are overlapped with previously reported psoriasis-associated genes or proteins. Red presents upregulation in the PE skin and green represents downregulation in the PE skin. Proteins of the 86 upregulated DEGs might be potential molecules involved in psoriasis pathogenesis in the PE skin. The DEGs are listed based on their molecule type.

**Figure 2 ijms-23-04983-f002:**
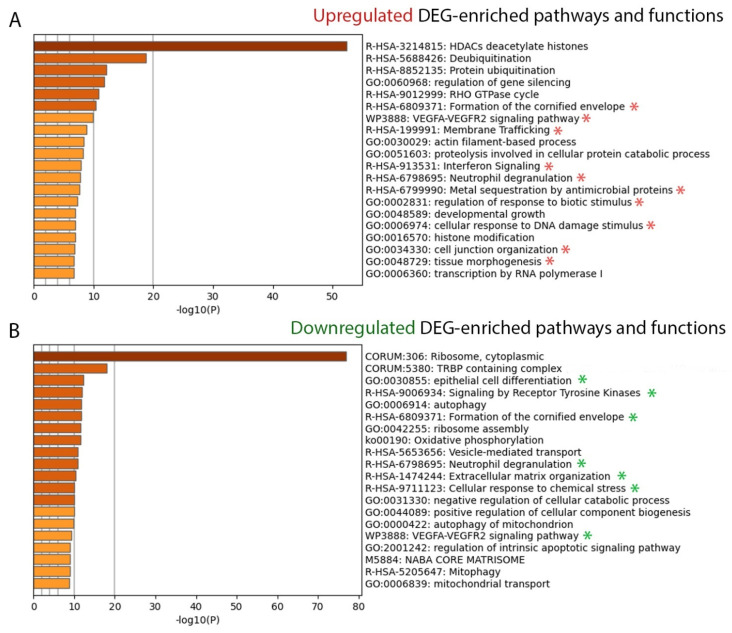
DEG-enriched pathway and function analysis (**A**) The top 20 upregulated DEG-enriched pathways and functions. (**B**) The top 20 downregulated DEG-enriched pathways and functions. A −log10 (10) value is the *p*-value in log base 10 to create the height of each bar chart. The associated pathways and functions are annotated with *. CORUM, comprehensive resource of mammalian protein complexes; DEG, differentially expressed gene; GO, gene ontology; ko, Kyoto Encyclopedia of Genes and Genomes (KEGG) pathway; M, canonical pathways; R-HSA, reactome gene sets; WP, WikiPathways.

**Figure 3 ijms-23-04983-f003:**
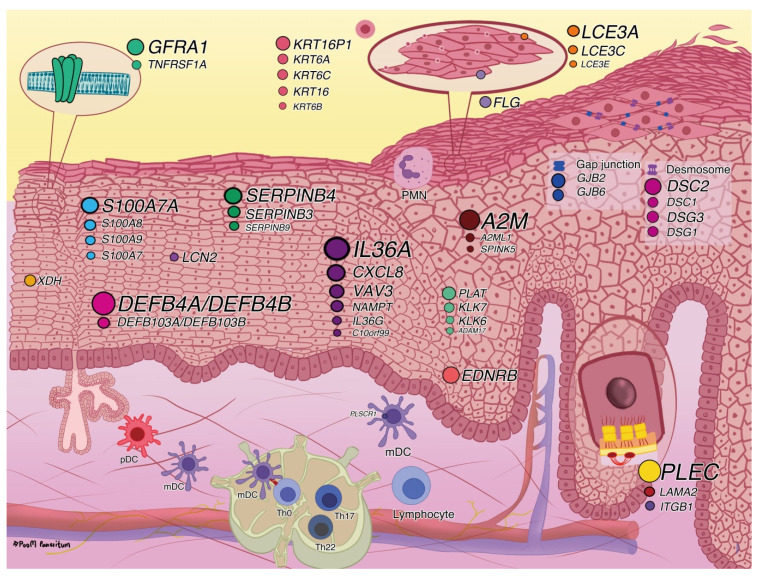
Potential upregulated differentially expressed genes (DEGs). The circles represent the fold change of expressed genes; the larger the circle, the more upregulated the expression of DEGs. *A2M*, alpha 2 macroglobulin; *A2ML1*, alpha 2 macroglobulin-like 1; *ADAM17*, ADAM metallopeptidase domain 17; *C10orf99*, chromosome 10 open reading frame 99; *CXCL8*, C-X-C motif chemokine ligand 8; *DEFB4A/DEFB4B*, defensin beta 4A; DSC2, desmocollin2; *DSG3*, desmoglein 3; *EDNRB*, endothelin receptor type B; *FLG*, filaggrin; *GFRA1*, GDNF family receptor alpha 1; *GJB2*, gap junction protein beta 2; *IL36A*, interleukin 36 alpha; *ITGB1*, integrin subunit beta 1; *KLK7*, kallikrein related peptidase 7; *KRT16P1*, keratin 16 pseudogene 1; *LAMA2*, laminin subunit alpha 2; *LCE3A*, late cornified envelope 3A; *LCN2*, lipocalin 2; mDC, myeloid dendritic cells; *NAMPT*, nicotinamide phosphoribosyltransferase; pDC, plasmacytoid dendritic cell; *PLAT*, plasminogen activator, tissue type; *PLEC*, plectin; *PLSCR1*, phospholipid scramblase 1; PMN, polymorphonuclear leukocytes; *S100A7A*, S100 calcium binding protein A7A; *SERPINB4*, serpin family B member 4; Th 0, T helper 0 cell; *TNFRSF1A*, TNF receptor superfamily member 1A; *VAV3*, vav guanine nucleotide exchange factor 3; *XDH*, xanthine dehydrogenase.

**Figure 4 ijms-23-04983-f004:**
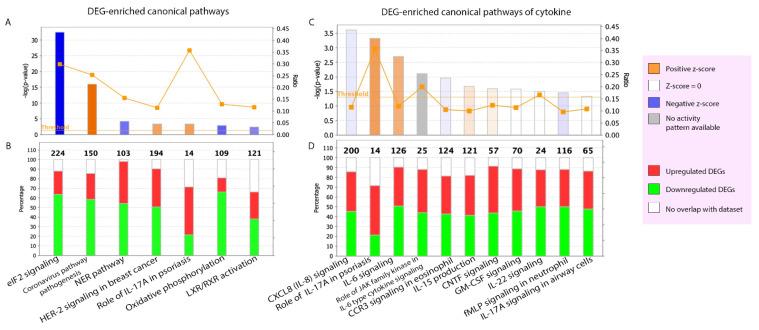
DEG-enriched canonical pathways. (**A**,**C**) The height of the bar represents the level of significance. The predicted activation pathways are shown by orange bars and the predicted inhibition pathways are shown by blue bars. The more intense the color, the stronger the activity prediction. The orange line connecting square points in each bar indicates the ratio of the dataset DEGs that are members of each known pathway divided by the total molecules’ pathway. (**B**,**D**) Percentage of DEGs overlapping in each pathway. Percentage of the non-overlapped pathway’s molecules is shown by the white segment of the bar. The total number of molecules in each canonical pathway are shown on the top of the bar. CCR3, C-C Motif Chemokine Receptor 3; CNTF, ciliary neurotrophic factor; CXCL8, C-X-C Motif Chemokine Ligand 8; DEG, differentially expressed gene; eIF2, eukaryotic initiation factor 2; fMLP, formyl-methionyl-leucyl-phenylalanine; GM-CSF, granulocyte-macrophage colony-stimulating factor; HER-2, human epidermal growth factor receptor; IL, interleukin; JAK, Janus Kinase; LXR/RXR, liver X receptor/retinoid X receptor; NER, Nucleotide excision repair.

**Figure 5 ijms-23-04983-f005:**
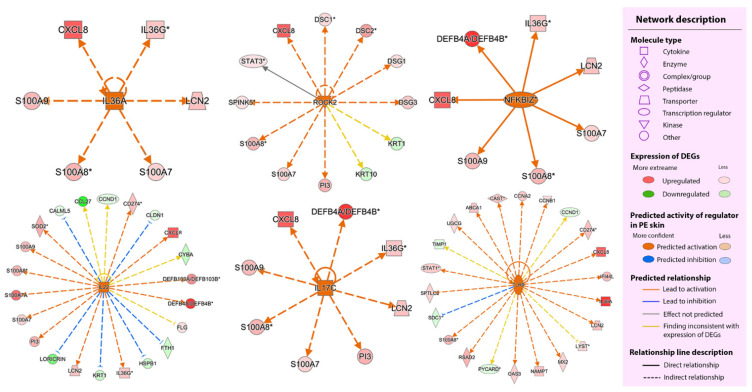
Potential regulators and their relationship to the target dataset’s DEGs. In each radiant, the molecule at the center is the predicted potential regulator and the outer circular molecules are the dataset’s DEGs. DEGs, differentially expressed genes; PE skin, peripheral edge of the lesional skin.

**Figure 6 ijms-23-04983-f006:**
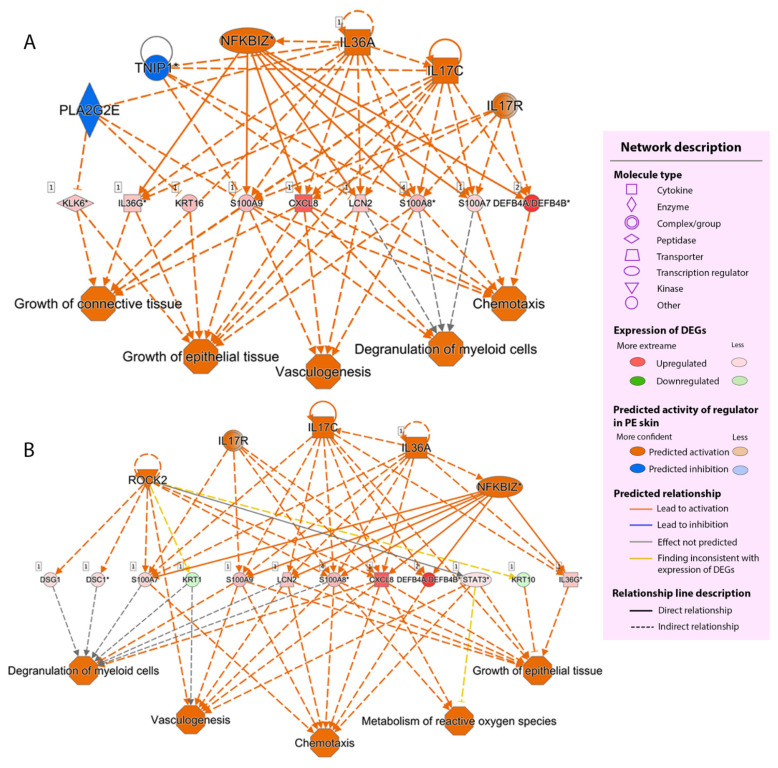
The two most useful regulator effect networks. (**A**) The highest consistency score network. (**B**) The second-highest consistency score network. Each network shows how predicted upstream regulators (top tier) control their dataset’s DEGs (middle tier) and the downstream functions of the dataset’s DEGs (bottom tier). DEGs, differentially expressed genes; PE skin, peripheral edge of the lesional skin.

**Figure 7 ijms-23-04983-f007:**
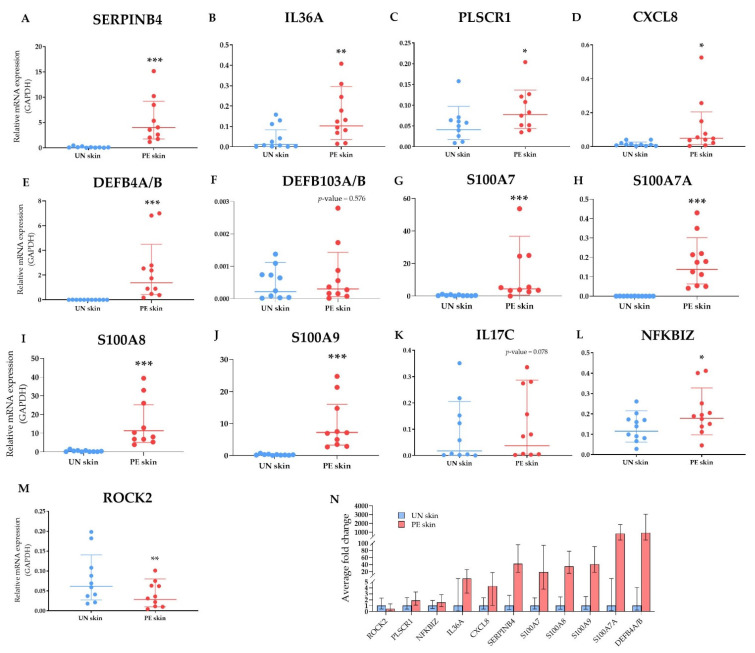
Quantitative multiplex Real-Time PCR analysis of DEGs and upstream regulators. The mRNA level of each gene was calculated relative to *GAPDH*, the housekeeping gene. (**A**–**J**) The relative RNA expression of selected DEGs. (**K**–**M**) The relative mRNA expression levels of selected upstream regulators. (**N**) The fold change of significantly altered genes. All graphs were designated by geometric mean and standard deviation. The significance was determined using *; * *p* < 0.05, ** *p* < 0.01 and *** *p* < 0.001. *n* ≥ 10 for each gene. CXCL8, C-X-C motif chemokine ligand 8; DEFB103A/B, defensin beta 103A/B; DEGs, differentially expressed genes; IL17C, interleukin 17C; NFKBIZ, nuclear factor kappa B inhibitor zeta; PE skin, peripheral edge of lesional skin; PLSCR1, phospholipid scramblase 1; ROCK2, rho-associated kinase 2; S100A7, S100 calcium binding protein A7; SERPINB4, serpin family B member 4; UN skin, uninvolved skin.

**Figure 8 ijms-23-04983-f008:**
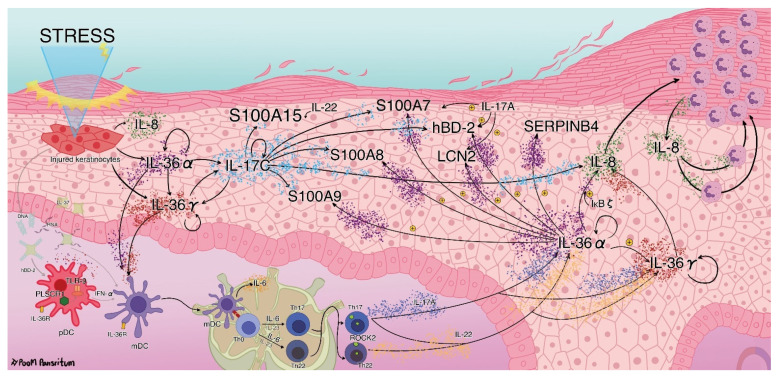
The proposed pathogenesis network at the peripheral edge (PE) of the lesional skin. Injured keratinocytes secrete interleukin (IL)-8, IL-36α, and IL-36γ. IL-36 activates plasmacytoid dendritic cells (pDCs) to secrete interferon-alpha (IFN)-α in a Toll-like receptor (TLR)-9-dependent manner (IL-36/TLR-9 axis). IL-36α and IL-36γ activate myeloid dendritic cells (mDCs) to mature and secrete IL-6, which drives the differentiation of T helper (h) cells (IL-6/Th17 axis). ROCK2 (rho-associated kinase 2) regulates CD4^+^ T cells to secrete IL-17. Note that inflammation in the PE skin is potentiated by IL-36α, IL-36γ, IL-17C, IL-8, S100A7, S100A8, S100A9, S100A15, SERPINB4, and hBD-2 and that a self-sustaining circuit between the innate and adaptive immune response in the PE skin is mediated through IL-36α and IL-36γ.

**Figure 9 ijms-23-04983-f009:**
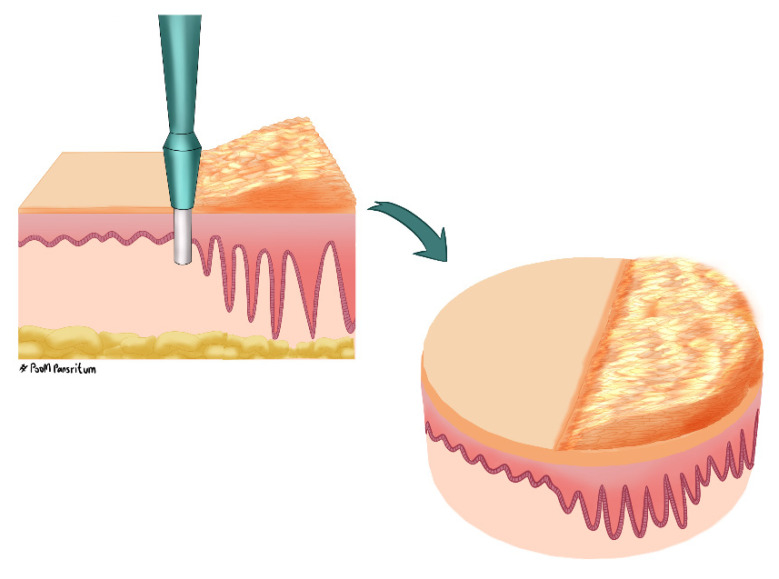
Peripheral edge of the lesional skin. The biopsy landmark was identified by the well-demarcated edge of the plaque. The piece of biopsy included equal sides of normal phenotypic skin and lesional skin.

**Table 1 ijms-23-04983-t001:** The top-50 differentially expressed genes of peripheral edge of the lesional skin compared with uninvolved skin.

	Symbol	Molecule Type *	Log 2FC	*p*-Value	Symbol	Molecule Type *	Log 2FC	*p*-Value
	**Upregulated DEGs**				**Downregulated DEGs**			
1	* A2M *	transporter	15.982	8.600 × 10^−6^	*ATP5F1A*	transporter	−27.343	7.380 × 10^−6^
2	*H3C2*	other	15.367	9.340 × 10^−5^	*SEPTIN5*	enzyme	−25.527	1.410 × 10^−3^
3	*PLEC*	other	15.295	1.360 × 10^−5^	*KHNYN*	other	−24.254	3.700 × 10^−3^
4	*ITSN2*	other	14.968	3.200 × 10^−4^	*PLD3*	enzyme	−23.010	5.630 × 10^−3^
5	*UGGT1*	enzyme	14.906	1.080 × 10^−4^	*FHL1*	other	−21.983	9.840 × 10^−3^
6	*STT3A*	enzyme	14.745	3.460 × 10^−3^	*HEXA*	enzyme	−20.826	8.320 × 10^−3^
7	* HERC6 *	enzyme	14.724	1.120 × 10^−3^	*DNM2*	enzyme	−19.584	7.500 × 10^−3^
8	* DEFB4A/DEFB4B *	other	14.690	1.400 × 10^−4^	*TLE5*	transcription regulator	−18.613	3.990 × 10^−3^
9	*CAND1*	transcription regulator	14.654	8.540 × 10^−3^	*MYH14*	enzyme	−16.846	1.460 × 10^−3^
10	* HRNR *	other	14.597	1.340 × 10^−5^	* GSN *	other	−16.812	1.490 × 10^−3^
11	* IL36A *	cytokine	14.572	1.340 × 10^−4^	*SPSB3*	other	−16.593	1.100 × 10^−3^
12	*SMARCA4*	transcription regulator	14.517	2.630 × 10^−5^	* LCE1C *	other	−16.153	1.620 × 10^−3^
13	*CYFIP1*	translation regulator	14.500	2.820 × 10^−3^	*LAMTOR4*	other	−16.058	9.610 × 10^−3^
14	*HIPK3*	kinase	14.365	6.200 × 10^−3^	*SYT8*	transporter	−16.014	9.260 × 10^−3^
15	*PLA2G4E-AS1*	other	14.274	8.590 × 10^−5^	*LTBP4*	growth factor	−15.985	2.600 × 10^−3^
16	*TRIP12*	enzyme	14.146	4.660 × 10^−3^	* ITGB4 *	transmembrane receptor	−15.717	3.270 × 10^−3^
17	*CLIC1*	ion channel	14.091	4.110 × 10^−5^	*DCTN1*	other	−15.591	3.700 × 10^−3^
18	*CD164*	other	14.069	1.280 × 10^−3^	*RTN4*	other	−15.532	3.930 × 10^−3^
19	*BZW1*	translation regulator	14.025	6.550 × 10^−3^	*CAMSAP3*	other	−15.517	3.990 × 10^−3^
20	*NRBP1*	kinase	13.939	1.280 × 10^−4^	*NFIX*	transcription regulator	−15.266	5.070 × 10^−3^
21	*PSMD12*	other	13.697	8.810 × 10^−5^	* CRAT *	enzyme	−15.226	4.210 × 10^−3^
22	*EPB41L3*	other	13.662	2.000 × 10^−3^	*RAB40C*	enzyme	−15.156	5.860 × 10^−3^
23	*SCYL2*	other	13.652	1.490 × 10^−4^	* CCL27 *	other	−15.021	3.820 × 10^−3^
24	*H2BC10*	other	13.610	1.870 × 10^−4^	*SIK1/SIK1B*	kinase	−14.994	6.560 × 10^−3^
25	*WDR75*	other	13.554	1.180 × 10^−3^	*CAPG*	other	−14.970	2.080 × 10^−3^
26	*H4C13*	other	13.550	1.080 × 10^−3^	*EPN2*	other	−14.879	7.780 × 10^−3^
27	*EXT2*	enzyme	13.432	4.740 × 10^−4^	*ARAP1*	other	−14.871	7.950 × 10^−3^
28	*H3C7*	other	13.405	1.480 × 10^−3^	*PDCD4*	other	−14.858	7.870 × 10^−3^
29	*PLEKHG5*	other	13.319	2.030 × 10^−4^	*XPO6*	other	−14.735	9.330 × 10^−3^
30	*DMXL1*	other	13.302	1.930 × 10^−3^	*PODN*	other	−14.705	9.560 × 10^−3^
31	*GBP6*	enzyme	13.278	2.320 × 10^−3^	*TUBGCP2*	peptidase	−14.694	9.730 × 10^−3^
32	*H3C11*	other	13.099	4.590 × 10^−4^	*CIRBP*	translation regulator	−14.658	9.860 × 10^−3^
33	*H2AC14*	other	13.092	4.200 × 10^−6^	*TSPAN14*	other	−14.640	9.940 × 10^−3^
34	*SMC6*	other	12.917	2.030 × 10^−4^	*ATP2B4*	transporter	−14.506	1.540 × 10^−3^
35	*H2AC21*	other	12.847	7.450 × 10^−4^	*CNOT1*	other	−14.414	8.590 × 10^−3^
36	*H2AC4*	other	12.793	5.310 × 10^−4^	*WBP2*	transcription regulator	−14.403	3.680 × 10^−3^
37	*CD59*	other	12.792	9.370 × 10^−4^	*MAN2C1*	enzyme	−14.324	2.770 × 10^−3^
38	*TMEM14EP*	other	12.753	6.240 × 10^−4^	*POLD2*	enzyme	−14.293	6.080 × 10^−3^
39	*GLB1*	enzyme	12.717	1.270 × 10^−3^	*MPST*	enzyme	−14.237	5.410 × 10^−3^
40	*CTNNBL1*	other	12.703	2.090 × 10^−3^	*ACSS2*	enzyme	−14.029	4.050 × 10^−3^
41	*GFPT1*	enzyme	12.637	3.910 × 10^−3^	*AP2A1*	transporter	−13.944	4.740 × 10^−3^
42	*H3C8*	other	12.599	1.300 × 10^−4^	*SCRIB*	other	−13.900	6.080 × 10^−3^
43	*ZDHHC6*	enzyme	12.565	4.940 × 10^−3^	*TAP2*	transporter	−13.808	8.590 × 10^−3^
44	*IKBKB*	kinase	12.437	2.280 × 10^−3^	*PHYHIP*	other	−13.783	6.750 × 10^−3^
45	*GANAB*	enzyme	12.412	2.170 × 10^−4^	* PLCH2 *	enzyme	−13.78	6.560 × 10^−3^
46	*TMEM184B*	other	12.410	1.410 × 10^−3^	*TMEM63B*	ion channel	−13.617	6.260 × 10^−3^
47	*SEPTIN11*	other	12.377	9.900 × 10^−4^	*TAX1BP1*	other	−13.600	3.020 × 10^−3^
48	*RMDN3*	other	12.322	3.100 × 10^−3^	* PKM *	kinase	−13.594	5.310 × 10^−3^
49	*LPP*	other	12.292	5.600 × 10^−3^	*EPHX3*	enzyme	−13.472	8.680 × 10^−3^
50	*H2BC13*	other	12.292	2.600 × 10^−3^	*PTBP1*	enzyme	−13.167	8.660 × 10^−3^

Based on QIAGEN Knowledge Base as of October 2021, the blue molecules have been previously mentioned in psoriasis, either their mRNAs or their proteins. * Molecule types are classified into complex, cytokine, enzyme, fusion gene/product, G-protein coupled receptor, group, growth factor, ion channel, kinase, ligand-dependent nuclear receptor, mature microRNA, peptidase, phosphatase, transcription regulator, translation regulator, transmembrane receptor and transporter. None of these were classified as other. FC, fold change.

**Table 2 ijms-23-04983-t002:** The potential differentially expressed genes (DEGs).

Gene Symbol	Log2FC	*p*-Value	Gene Symbol	Log2FC	*p*-Value
**Upregulation**			**Downregulation**		
**Anti-microbial peptides (AMPs)**			**S100 proteins**		
*DEFB4A/DEFB4B*	14.690	1.400 × 10^−4^	***S100A16***	−5.334	9.920 × 10^−3^
*DEFB103A/DEFB103B*	7.624	2.290 × 10^−4^	***S100A10***	−4.100	2.780 × 10^−5^
*S100A7A (S100A15*)	9.559	1.180 × 10^−5^	***S100A4***	−3.819	1.450 × 10^−4^
*S100A8*	4.944	2.590 × 10^−4^	**Cytokines**		
*S100A9*	4.722	3.440 × 10^−5^	***FBRS***	−3.204	2.140 × 10^−3^
*S100A7*	3.002	3.930 × 10^−5^	***SLURP1***	−1.975	3.420 × 10^−3^
*LCN2*	4.182	8.240 × 10^−4^	***TIMP1***	−1.857	4.290 × 10^−3^
**Cytokines**			*MIF*	−1.842	4.860 × 10^−3^
*IL36A*	14.572	1.340 × 10^−4^	***CXCL14***	−1.752	1.650 × 10^−3^
*CXCL8*	10.681	6.580 × 10^−4^	**Transmembrane receptor**		
***VAV3***	9.178	1.290 × 10^−4^	*ITGB4*	−15.717	3.270 × 10^−3^
*NAMPT*	3.919	8.090 × 10^−4^	**Cell adhesion molecules**		
*IL36G*	3.897	2.090 × 10^−3^	***GJB3***	−5.454	9.220 × 10^−3^
*C10orf99*	2.567	2.490 × 10^−3^	***GJB5***	−2.317	9.110 × 10^−3^
**Transmembrane receptors**			*LAMA5*	−2.125	7.100 × 10^−3^
***GFRA1***	9.023	5.660 × 10^−3^	**Keratin and late cornified envelope**		
***TNFRSF1A***	3.317	1.180 × 10^−3^	*KRT2*	−5.484	9.530 × 10^−4^
***ITGB1***	2.030	8.210 × 10^−3^	*KRT1*	−3.700	3.250 × 10^−4^
**Cell adhesion molecules**			*KRT10*	−3.521	1.510 × 10^−4^
*DSC2*	6.595	1.260 × 10^−4^	***KRT77***	−3.416	4.850 × 10^−4^
***DSC1***	2.478	6.390 × 10^−3^	*LCE1C*	−16.153	1.620 × 10^−3^
*DSG3*	3.599	1.590 × 10^−3^	***LCE2C/LCE2D***	−5.366	9.450 × 10^−4^
*DSG1*	2.456	7.280 × 10^−3^	*LCE2B*	−4.689	6.280 × 10^−5^
*GJB2*	5.468	3.430 × 10^−6^	***LCE1B***	−3.871	8.320 × 10^−5^
***GJB6***	5.135	4.940 × 10^−3^	***LCE6A***	−3.563	5.910 × 10^−4^
***PLEC***	15.295	1.360 × 10^−5^	***LCE2A***	−3.467	1.230 × 10^−4^
*LAMA2*	1.626	8.630 × 10^−3^	***LCE1D***	−3.371	9.840 × 10^−4^
**Keratin and late cornified envelope**			***LCE1A***	−3.132	1.310 × 10^−3^
***KRT16P1***	8.120	8.130 × 10^−3^	***LCE1F***	−2.956	1.890 × 10^−3^
*KRT6A*	5.740	3.170 × 10^−4^	**Peptidase**		
*KRT6C*	5.310	1.170 × 10^−4^	*KLK11*	−5.992	2.640 × 10^−5^
*KRT16*	4.675	8.870 × 10^−5^	**Other interesting DEGs**		
*KRT6B*	3.384	2.000 × 10^−3^	*APOE*	−3.300	6.710 × 10^−3^
***LCE3A***	7.383	2.430 × 10^−5^	*CALML5*	−2.153	6.690 × 10^−4^
*LCE3C*	5.451	1.000 × 10^−3^	*CAV1*	−2.121	5.430 × 10^−3^
*LCE3E*	2.652	2.210 × 10^−3^	*CCL27*	−15.021	3.820 × 10^−3^
**Peptidase**			*CCND1*	−1.827	4.370 × 10^−3^
*PLAT*	4.388	5.750 × 10^−4^	***CD81***	−3.603	1.190 × 10^−4^
*KLK7*	4.224	7.300 × 10^−4^	***COL16A1***	−2.658	7.960 × 10^−4^
*KLK6*	3.927	2.030 × 10^−3^	***COL1A1***	−2.636	1.870 × 10^−4^
*ADAM17*	2.146	3.400 × 10^−3^	***COL6A1***	−3.547	8.270 × 10^−5^
**Protease inhibitor**			***COL6A2***	−3.357	3.740 × 10^−3^
*A2M*	15.982	8.600 × 10^−6^	***COL7A1***	−1.766	1.970 × 10^−3^
***A2ML1***	1.703	7.220 × 10^−3^	***FBL***	−3.122	4.290 × 10^−3^
***SPINK5***	1.574	4.340 × 10^−3^	*GSN*	−16.812	1.490 × 10^−3^
**Other interesting DEGs**			***GPX4***	−3.137	4.430 × 10^−3^
***CD59***	12.792	9.370 × 10^−3^	***SERPINF1***	−2.115	3.820 × 10^−3^
***IKBKB***	12.437	2.280 × 10^−3^	*IL20RB*	−1.788	2.680 × 10^−3^
***EDNRB***	11.815	5.820 × 10^−3^	***IGFBP6***	−5.399	1.260 × 10^−3^
*TNIP3*	10.794	6.580 × 10^−3^	***IGFBP4***	−3.248	1.140 × 10^−4^
*SERPINB4*	10.691	7.430 × 10^−3^	*IGFBP7*	−1.574	6.940 × 10^−3^
*SERPINB3*	7.630	3.290 × 10^−5^	***ITGB5***	−2.077	7.180 × 10^−3^
***SERPINB9***	3.960	6.270 × 10^−3^	***LGALS7/LGALS7B***	−4.880	8.880 × 10^−5^
***XDH***	5.565	6.140 × 10^−3^	*LGALS1*	−3.681	1.030 × 10^−3^
*PI3*	5.139	5.430 × 10^−5^	*LORICRIN*	−9.244	8.540 × 10^−4^
*STAT1*	3.384	3.940 × 10^−3^	***MMP28***	−11.574	7.540 × 10^−3^
*STAT3*	1.617	8.030 × 10^−3^	*NUPR1*	−2.183	3.910 × 10^−3^
***DDX21***	2.520	2.580 × 10^−3^	***PECAM1***	−2.517	2.980 × 10^−3^
*PLSCR1*	2.352	3.520 × 10^−3^	*PSORS1C2*	−3.373	8.360 × 10^−3^
***RICTOR***	2.338	8.330 × 10^−3^	*TIMP2*	−3.092	3.820 × 10^−4^
*FLG*	2.135	6.550 × 10^−4^	*TNXB*	−2.238	2.460 × 10^−3^
***KDM5A***	1.905	2.380 × 10^−3^	*TYK2*	−1.915	5.780 × 10^−3^
***HLA-A***	1.825	1.170 × 10^−3^	***VEGFB***	−3.382	6.940 × 10^−3^
*IGFBP3*	1.600	1.200 × 10^−3^	*WNT7B*	−3.574	4.310 × 10^−3^

These DEGs were selected due to their biofunction and psoriasis association. Based on QIAGEN Knowledge Base as of October 2021, ***the bold molecules*** have not been previously mentioned in psoriasis. FC, fold change.

## Data Availability

The raw data generated in this study have been deposited in the NCBI Gene Expression Omnibus (GEO; http://www.ncbi.nlm.nih.gov/geo (accessed on 1 March 2022)) and are accessible through the GEO Series accession number GSE183732.

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
