# Peer review of "Transcriptomic Profiling of Peripheral Edge of Lesions to Elucidate the Pathogenesis of Psoriasis Vulgaris"

_ijms, 2022, doi:10.3390/ijms23094983_

Round 1
Reviewer 1 Report
The paper already underwent different rounds of revisions. i think that provides new insights into signaling pathways in peripheral edge skin, which could lead to the discover of new drugs. I would probably add some more pieces of information in the introduction about currently available biological drugs for psoriasis (here an interesting article:doi: 10.1371/journal.pone.0241575. )
Still, the paper is in my opinion eligible to be published after this minor correction
Author Response
Response to reviewer 1
We are truly grateful for your positive comments and valuable suggestions on our work. We have tried our best to improve the manuscript according to your suggestions. Attached below are point-by-point responses to each of the comments raised by the reviewer.
The paper already underwent different rounds of revisions. I think that it provides new insights into signaling pathways in peripheral edge skin, which could lead to the discover of new drugs. I would probably add some more pieces of information in the introduction about currently available biological drugs for psoriasis (here an interesting article: Doi: 10.1371/journal.pone.0241575. )
Response
We have added a passage to the introduction section of the manuscript detailing this, as well as an appropriate citation to the reference mentioned by the reviewer. (Line 60–63)

Reviewer 2 Report
A very interesting original article identifing a new set of skin genes and signaling pathways in peripheral edge of the lesional skin in psoriatic patients that help to generate the proposed pathogenesis in the peripheral edge of the lesional skin. These could lead to a discovery of new therapeutic targets in psoriasis vulgaris. Multiple rounds of revisions were already performed, and altough the limited number of patients may represent a problem and the main limitation for this study, i think it will be eligible for publication after minor adjustments:
line 37 you should add: "Different clinical phenotypes of psoriasis have been reported, including palmoplantar, inverse, guttate, pustular, and others. It may be associated with comorbidities, such as arthritis." and cite: doi: 10.3390/pharmaceutics14020294. and doi: 10.3390/healthcare9050543.
Conclusions should be expanded, also discussing the future perspectives following this study.
Author Response
Response to reviewer 2
We are truly grateful for your positive comments and valuable suggestions on our work. We have tried our best to improve the manuscript according to your suggestions. Attached below are point-by-point responses to each of the comments raised by the reviewer.
1) line 37 you should add: "Different clinical phenotypes of psoriasis have been reported, including palmoplantar, inverse, guttate, pustular, and others. It may be associated with comorbidities, such as arthritis." and cite: doi: 10.3390/pharmaceutics14020294. and doi: 10.3390/healthcare9050543.
Response
We have added a passage describing this to the Introduction section of the manuscript and added an appropriate citation to the reference mentioned by the reviewer. (Line 37–40)
2) Conclusions should be expanded, also discussing the future perspectives following this study.
Response
We have expanded the conclusions of the manuscript, adding details regarding future perspectives and limitations of the study. (Line 400–415)

This manuscript is a resubmission of an earlier submission. The following is a list of the peer review reports and author responses from that submission.